# Association between Brachial-Ankle Pulse Wave Velocity and Microalbuminuria and to Predict the Risk for the Development of Microalbuminuria Using Brachial-Ankle Pulse Wave Velocity Measurement in Type 2 Diabetes Mellitus Patients

**DOI:** 10.3390/healthcare7040111

**Published:** 2019-09-26

**Authors:** Byong-Kyu Kim, Dilaram Acharya, Deuk-Young Nah, Moo-Yong Rhee, Seok-Ju Yoo, Kwan Lee

**Affiliations:** 1Cardiology Division, Department of Internal Medicine, Gyeongju Hospital, College of Medicine, Dongguk University, 87 Dongdaero, Gyeongju, Gyeongbuk 38067, Korea; bleumatin@dongguk.ac.kr; 2Department of Preventive Medicine, College of Medicine, Dongguk University, Gyeongju 38066, Korea; dilaramacharya123@gmail.com (D.A.); medhippo@hanmail.net (S.-J.Y.); kwaniya@dongguk.ac.kr (K.L.); 3Department of Community Medicine, Kathmandu University, Devdaha Medical College and Research Institute, Rupandehi 32907, Nepal; 4Cardiovascular Center, Dongguk University Ilsan Hospital, 27, Dongguk-ro, Ilsandong-gu, Goyang-si, Gyeonggi-do 10326, Korea; mooyong.rhee@dumc.or.kr

**Keywords:** diabetes mellitus, microalbuminuria, pulse wave velocity

## Abstract

Brachial-ankle pulse wave velocity (baPWV) provides a useful means of assessing cardiovascular events and diabetic complications. However, the nature of associations between baPWV and microalbuminuria (MAU) and its presence in Type 2 diabetes mellitus (Type 2 DM) have rarely been investigated. This study aimed to examine the association between baPWV and MAU coupled with prediction of MAU using baPWV measurement among Type 2 DM patients. In this cross-sectional study, we enrolled 424 Type 2 DM patients who visited the cardiology and endocrinology department at a tertiary level health care facility, Republic of Korea between 1 January 2006 to 31 December 2008. Clinical and laboratory data were collected, and risk factors associated with MAU and prediction of risk for the development of MAU using baPWV measurement. The association between MAU and baPWV was examined using multivariable logistic regression analysis and predicted MAU by using receiver operating characteristic (ROC) curve analysis. Of the 424 Type 2 DM patients, 93 (21.9%) had MAU (20–200 μg/min). baPWV (cm/sec) was found to be significantly correlated with MAU levels (ug/min) (*r* = 0.791, *p* < 0.001). Further, baPWV was significantly associated MAU with higher odds ratio (adjusted odds ratio (AOR) 10.899; 95% confidence interval (CI) (4.518–26.292)). Similarly, smoking (AOR 5.736; 95% CI (1.036–31.755)), and low-density lipoprotein (LDL)-cholesterol (mg/dL) (AOR 1.017; 95% CI (1.001–1.033)) were also significantly associated with MAU. The appropriate cut-off value for baPWV to predict MAU 20 μg/min in our study was 1700 cm/sec (area under ROC curve = 0.976). This study shows that baPWV, cigarette smoking, and LDL-cholesterol are associated with MAU in Type 2 DM patients and suggests that a baPWV cut-off of 1700 cm/sec could be used to predict the presence of MAU (20 μg/min) in Type 2 DM patients in the Korean community.

## 1. Introduction

Diabetes mellitus is a global public health problem. In 2016, an estimated 1.6 million deaths were caused by diabetes and the number affected is predicted to increase to 552 million by 2030 [1,2]. In addition to increasing morbidity and mortality, Type 2 diabetes mellitus (Type 2 DM) is considered to be an independent risk factor of many cardiovascular illnesses, such as coronary artery disease, and peripheral artery disease [3,4,5]. On the other hand, microalbuminuria (MAU) is a clinical condition that can lead to proteinuria and cause endothelial dysfunction and atherosclerosis, and these conditions can increase arterial stiffness in Type 2 DM [6,7]. MAU has been reported to be positively associated with increased risks of cardiovascular morbidity and mortality independently of conventional cardiovascular risk factors in Type 2 DM [8]. Interestingly, MAU has also been reported to be an important risk factor for the development of cardiovascular diseases and for increased mortality among the general population, and even at borderline level of microalbumin in general populations are reported to be significantly increased risk of death [9]. However, the pathogenic mechanism of MAU is not well understood. One theory is that insulin resistance is common among patients with diabetes or dyslipidemia and results from endothelial dysfunction triggered by metabolic processes [10,11,12], while another posits that generalized vascular dysfunction due to arterial stiffness may result in MAU [13,14].

Arterial stiffness has been reported to be an important marker of vascular damage and a risk factor of cardiovascular diseases [10,12] and albuminuria [15], and increased pulse wave velocity (PWV) is known to be an indicator of arterial stiffness. Furthermore, a straightforward method of measuring brachial-ankle pulse wave velocity (baPWV) is now available, although it should be added that carotid-femoral pulse wave velocity (cfPWV) is the established standard method for measuring PWV [16]. However, there has been documented evidence of close correlation between baPWV and cfPWV, with similar determinants. As such, baPWV measurement may be used as an alternative to cfPWV measurement in order to investigate arterial stiffness for assessing cardiovascular risk [17]. baPWV reflects the stiffness of the aorta and peripheral arteries in lower and upper extremities, and is more applicable in general practice than cfPWV, as it uses a separate cuff for each limb, is automated, and easier to perform [18,19,20].

Several authors have concluded baPWV provides a useful means of quantitatively assessing cardiovascular events and diabetic complications [21,22,23], and cross-sectional and longitudinal studies have shown baPWV is independently associated with MAU [24,25,26]. Others have suggested baPWV cut-off values for the prediction of mortality [18] and the development of cardiovascular illnesses [27]. However, few Korean studies have addressed the nature of the association between MAU and baPWV in Type 2 DM and the influences of other risk factors on this association, but did not examine the ability of baPWV to predict the presence of MAU in Type 2 DM. Although MAU can easily be performed in clinic, we hypothesized that an elevated baPWV, if found positively associated with MAU, may be a good indicator for microalbuminuria, and also it could be a useful screening tool to predict cardiovascular complications among Type 2 DM patients. Given this background, the current study was conducted to examine the association between baPWV and MAU coupled with prediction of MAU using baPWV measurement among Type 2 DM patients.

## 2. Materials and Methods

### 2.1. Study Subjects

Four hundred and twenty-four consecutive Type 2 DM patients who visited the department of cardiology and endocrinology of Dongguk University, Gyeongju Hospital from 1 January 2006 to 31 December 2008 were enrolled in this cross-sectional study. All subjects underwent baPWV and MAU testing. Patients with renal insufficiency (serum creatinine ≥1.5 mg/dl) or clinical albuminuria (≥200 μg/min) were excluded. The 424 study subjects were allocated to one of two groups based on microalbumin level, that is, to a normoalbuminuria (NAU) group (<20 μg/min (*n* = 331)) or a microalbuminuria (MAU) group (20–200 μg/min (*n* = 93)), based on available literature [28].

### 2.2. Measurement of Pulse Wave Velocity

baPWV was measured in a quiet room controlled at 22 ± 1 °C in an overnight fasted state. All study subjects were asked to refrain from caffeine, alcohol, and smoking during the 12 h period before testing. After having a 15 min rest, baPWV was measured using an automated device (VP-1000; Colin, Co. Ltd, Komaki, Japan) in the supine position [29]. Briefly, pressure waveforms of brachial and tibial arteries were obtained using occluding monitoring cuffs placed around upper arms and lower legs. Times taken for pulse waves to travel from lower legs to upper arms were recorded; distances between sampling points of baPWV were calculated automatically from subject heights. One technician performed all measurements.

### 2.3. Ethics

The study protocol was approved by the institutional review board of Dongguk University Gyeongju Hospital (approval number: 110757-201712-HR-03-01). Subject privacy, confidentiality, and anonymity were fully maintained throughout the study. A written informed consent was taken from each of the study participants after giving information about study details.

### 2.4. Physical and Laboratory Measurements

Physical measurements, such as heights and weights, were measured on the same days as baPWV tests. Similarly, hypertensions, hypercholesterolemia, smoking status, and diabetes duration were obtained at these times. Lipid profiles, which included total cholesterol, triglyceride, low-density lipoprotein cholesterol, high-density lipoprotein cholesterol, blood sugar level, hemoglobin A1C, and uric acid, were measured enzymatically. All blood samples were obtained in the morning in an overnight fasted state. Urinary microalbumin concentrations were measured using a radioimmunoassay (Dream r-10 counter, Shinjin Medics, Seoul, South Korea) according to the manufacturer’s instructions.

### 2.5. Statistical Analysis

Demographic and clinical characteristics are presented as means and standard deviations (SDs) or as numbers and percentages, and were compared using the independent t-test or the Pearson’s chi-square test depending on data type. Pearson’s correlation analysis was used to identify the relationship between baPWV and the presence of MAU. All variables with a *p*-value of ≤0.1 were entered the multivariable logistic regression analysis with backward elimination. Adjusted odds ratios and 95% confidence intervals (CIs) were calculated to determine the independence of the association between MAU and baPWV. *p*-values < 0.05 were deemed statistically significant, and the analysis was conducted using SPSS version 20 (SPSS, Chicago, IL, USA). In addition, receiver operating characteristic (ROC) analysis in MedCalc Statistical Software version 17.9.7 was used to determine the optimal baPWV cut-off for predicting the presence of MAU [30].

## 3. Results

### Baseline and Clinical Characteristics of the Study Subjects

The baseline characteristics and laboratory findings of the study subjects are summarized in Table 1 and Table 2. Of the 424 Type 2 DM patients, 93 (21.9%) had microalbuminuria. Study subjects were allocated to a normoalbuminuria (NAU) group (*n* = 331, <20 μg/min) or a microalbuminuria (MAU) group (*n* = 93, 20–199 μg/min). Smoking (14.2% vs. 26.9%, *p* = 0.004), duration of diabetes (7.5 ± 5.7 vs. 10.4 ± 8.8 years, *p* = 0.001), and weight (63.7 ± 11.3 vs. 66.6 ± 12.8 kg, *p* = 0.036) were significantly different in the NAU and MAU groups (Table 1). Similarly, blood urea nitrogen (BUN) (16.3 ± 5.1vs. 19.3 ± 8.8 mg/dl, *p* = 0.002), microalbumin (6.5 ± 4.8 vs. 66.9 ± 43.0 ug/min, *p* = 0.001), and baPWV (1,459.15 ± 200.44 vs. 1,981.74 ± 171.47 cm/sec, *p* = 0.001) were also significantly different in the NAU and MAU groups (Table 2).

In all study subjects, baPWV (cm/sec) was significantly correlated with microalbumin levels (ug/min) (*r* = 0.791, *p* < 0.001) (Figure 1).

Multivariable logistic regression analysis results with adjusted odds ratios and 95% confidence intervals (CIs) are shown in Table 3. baPWV was significantly associated with the MAU with higher odds ratio (AOR 10.899; 95% CI (4.518–26.292)), and, similarly, smokers were more likely (AOR 5.736; 95% CI (1.036–31.755)) to have MAU than nonsmokers, and those with a high LDL-cholesterol (mg/dL) were also found to be more likely (AOR 1.017; 95% CI (1.001–1.033)) to have MAU.

The receiver operating characteristic (ROC) curve analysis revealed that the optimum baPWV cut-off for predicting the presence of MAU (20 μg/min) was 1700 cm/sec (95.7% sensitivity, 89.7% specificity, and area under the curve (AUC) = 0.976)) (Figure 2).

## 4. Discussion

The principal finding of the present study was that brachial-ankle pulse wave velocity was significantly positively associated with microalbuminuria (MAU) in Type 2 DM. In addition, smoking and a high LDL-cholesterol were also significantly positively associated with MAU. More importantly, receiver operating characteristic (ROC) curve analysis revealed that the optimum baPWV cut-off for predicting the presence of MAU (20 μg/min) in Type 2 DM was 1700 cm/sec.

This study showed that 21.9% of the 424 Type 2 DM patients included had MAU, which is much higher than the rates found in previous Korean studies [26,31]. However, previous Korean studies were conducted on healthy subjects. In addition, univariate and bivariate analyses showed that baPWV and MAU were significantly positively associated in our cohort, which is in-line with the findings of previous studies conducted in China, Japan, and South Korea [25,26,32,33]. Furthermore, this significant association remained consistent with a high odds ratio even after controlling for potential confounders. Although not precisely determined, endothelial damage is probably a major cause of MAU [10,34]. For instance, MAU becomes fixed when progression of vascular structural changes starts including glomerulosclerosis, and endothelial dysfunction could lead to the development of MAU [10], which supports the notion that MAU reflects vascular damage in diabetic patients [35,36]. Interestingly, in a large community-based Japanese study, MAU was found to be independently associated with baPWV in the general population [25], and the results obtained suggested that the general population might have been suffering from MAU.

Since MAU is linked to kidney damage and chronic kidney disease, it may be an important marker of renal and cardiovascular illnesses [37,38,39]. Thus, the detection of MAU among Type 2 DM patients might provide a straightforward means of identifying patients at risk and of facilitating early treatment. Kawai et al. [27] reported that a baPWV cut-off of 1750 cm/sec might be useful for predicting the presence of cardiovascular disease and stroke in hypertensive patients, and Wang et al. suggested that a combination of baPWV > 1400 cm/sec and a retinal artery atherosclerosis ≥ 2 might usefully predict the presence of coronary artery disease [40]. Although these studies provided useful information regarding the detection of cardiovascular illnesses using baPWV measurements, they did not address the predictive value of baPWV for MAU in Type 2 DM. Nonetheless, despite inter-study differences, our results are in-line with previous reports. In contrast to previous studies [21,22,26,32,41], we used receiver operating characteristic (ROC) curve analysis to identify an optimal baPWV cut-off for the detection of MAU (20 μg/min), which was found to be 1700 cm/sec (95.7% sensitivity, 89.7% specificity, and AUC 0.976).

We also observed conventional risk factors of cardiovascular and renal illnesses, such as cigarette smoking and higher LDL-cholesterol, were significantly associated with MAU. Cigarette smoking has been consistently reported to be associated with MAU in longitudinal and cross-sectional studies [42,43], and, likewise, elevated levels of higher LDL-cholesterol have been reported to be associated with MAU [44]. It has been demonstrated that smoking alters glomerular filtration rate and increases urinary albumin:creatinine ratio among Type 2 DM patients [45]. On the other hand, dyslipidemia is associated with the atherosclerotic process, and recent research has shown total cholesterol independently predicts coronary artery calcification and incidental albuminuria [46,47,48].

Despite its usefulness, we acknowledge that the present study has some important limitations. First, its cross-sectional nature and relatively small sample size prevent us from determining cause and effect relationships. Second, our study finding whereby baPWV cut-off of 1700 cm/sec predicted the presence of MAU in Type 2 DM patients in a Korean community may be not be applicable to other races. Third, our study lacks some important information on some variables, such as different cardiovascular diagnoses and correlation between cfpWV and baPWV. Fourth, study subjects were receiving hypoglycemic and antihypertensive medication, and this may have influenced our findings. Nevertheless, the association found between baPWV and MAU might indicate incipient nephropathy or cardiovascular illness in some Type 2 DM patients [6,7], and that it might be beneficial to bring this segment of the population in for appropriate medical intervention when a simple and useful measurement is known.

## 5. Conclusions

In the present study, baPWV, cigarette smoking, and a high LDL-cholesterol level were found to be associated with MAU among Type 2 DM patients. In addition, the study indicates a baPWV value of 1700 cm/sec might be an important cut-off value for predicting the presence of MAU (20 μg/min) in Type 2 DM patients in the Korean community. Further, longitudinal, large-scale, multicenter studies are required to investigate the association between baPWV and microalbuminuria, pulse wave velocity (PWV), carotid intima-media thickness (IMT), and ambulatory blood pressure (ABP) in Korean Type 2 DM patients.

## Figures and Tables

**Figure 1 healthcare-07-00111-f001:**
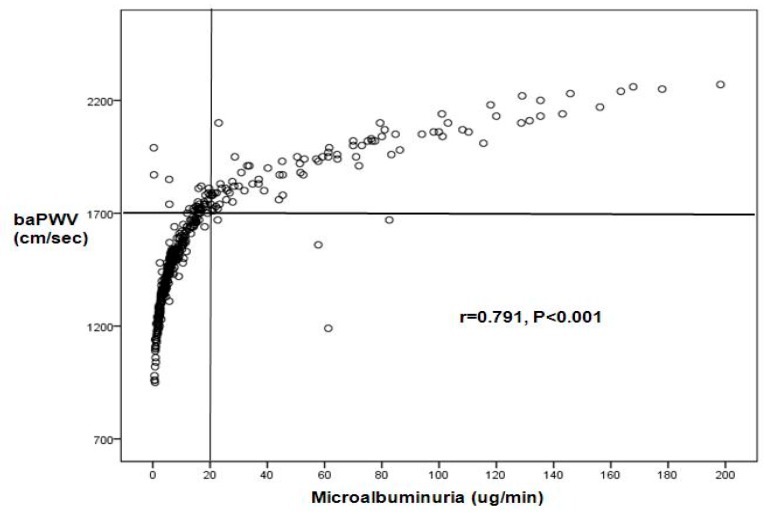
The relationship between microalbuminuria (MAU) and brachial-ankle pulse wave velocity (baPWV).

**Figure 2 healthcare-07-00111-f002:**
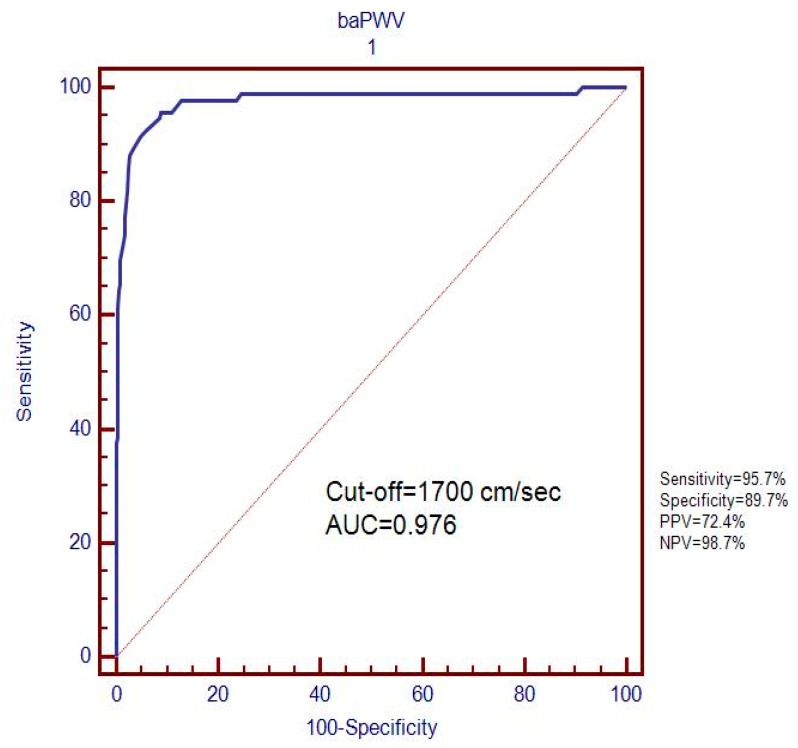
Determination of the optimum baPWV value for predicting the presence of microalbuminuria (20 μg/min).

**Table 1 healthcare-07-00111-t001:** Baseline clinical characteristics of the study subjects.

Variables	Normoalbuminuria (*n* = 331) (%)	Microalbuminuria (*n* = 93) (%)	*p*-Value
Male, n (%)	173 (52.3)	57 (61.3)	0.077
Age, years	58.2 ± 12.2	59.5 ± 13.0	0.369
HTN, n (%)	124 (37.5)	37 (38.7)	0.385
H-Chol, n (%)	141 (42.6)	36 (38.7)	0.291
Smoking, n (%)	47 (14.2)	25 (26.9)	0.004
DM duration, Years	7.5 ± 5.7	10.4 ± 8.8	0.001
Height (cm)	161 ± 9	163 ± 10	0.073
Weight (kg)	63.7 ± 11.3	66.6 ± 12.8	0.036
BMI	24.5 ± 3.9	24.8 ± 3.3	0.389
SBP, mmHg	125 ± 16	125 ± 18	0.956
DBP, mmHg	77 ± 10	76 ± 11	0.522
PP, mmHg	47 ± 10	48±11	0.53

HTN, hypertension; H-Chol, hypercholesterolemia; DM, diabetes mellitus; cm, centimeter; kg, kilogram, BMI; body mass index; SBP, systolic blood pressure; DBP, diastolic blood pressure; PP, pulse pressure.

**Table 2 healthcare-07-00111-t002:** Clinical characteristics of the study subjects.

Variables	Normoalbuminuria (*n* = 331) (%)	Microalbuminuria (*n* = 93) (%)	*p*-Value
FBS (mg/dl)	136 ± 54	138 ± 48	0.722
HbA1c (%)	7.7 ± 2.0	8.1 ± 2.1	0.107
BUN (mg/dL)	16.3 ± 5.1	19.3 ± 8.8	0.002
Creatinine (mg/dL)	0.99 ± 0.18	1.14 ± 0.72	0.05
Total cholesterol (mg/dL)	179 ± 41	184 ± 45	0.369
Triglyceride (mg/dL)	160 ± 113	158 ± 78	0.864
HDL-cholesterol (mg/dL)	46.3 ± 13.1	45.4 ± 12.7	0.551
LDL-cholesterol (mg/dL)	103 ± 33	109 ± 40	0.138
Uric Acid	4.9 ± 1.4	5.2 ± 1.8	0.154
Microalbumin (μg/min)	6.5 ± 4.8	66.9 ± 43.0	0.001
baPWV (cm/sec)	1459.15 ± 200.44	1981.74 ± 171.47	0.001

FBS, fasting blood sugar; HbA1c, hemoglobin A1c; BUN, blood urea nitrogen; HDL, high-density lipoprotein; LDL, low-density lipoprotein; baPWV, brachial-ankle pulse wave velocity.

**Table 3 healthcare-07-00111-t003:** Association between brachial-ankle pulse wave velocity and microalbuminuria as determined by multivariable logistic regression analysis **.

Variable	Regression Coefficient	SE	AOR	95% CI	*p*-Value
baPWV *	2.389	0.449	10.899	4.518–26.292	0.0001
Smoking					
No	-	-	Ref [1]	-	-
Yes	1.747	0.873	5.736	1.036–31.755	0.045
Creatinine (mg/dl) *	1.909	0.984	6.745	0.980-46.432	0.052
LDL-cholesterol (mg/dl) *	0.017	0.008	1.017	1.001–1.033	0.035

baPWV, brachial-ankle pulse wave velocity; SE, standard error; AOR, adjusted odds ratio; CI, confidence interval; Ref, reference. * continuous variable. ** Adjusted for gender, smoking, diabetes mellitus duration, height, weight, blood urea nitrogen, creatinine, low-density lipoprotein, uric acid, microalbuminuria, and baPWV.

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
