# Peer review of "Association between Brachial-Ankle Pulse Wave Velocity and Microalbuminuria and to Predict the Risk for the Development of Microalbuminuria Using Brachial-Ankle Pulse Wave Velocity Measurement in Type 2 Diabetes Mellitus Patients"

_healthcare, 2019, doi:10.3390/healthcare7040111_

Round 1
Reviewer 1 Report
This study revealed an association between baPWV and MAU and a cut-off value of baPWV to predict MAU in a cohort of T2DM patients. Below are my comments
The introduction highlights the importance of baPWV and MAU in predicting cardiovascular disease, but it is not clear why it is important to predict MAU using baPWV. Please clarify. MAU can easily be performed in the clinic. What is the reason for using baPWV to predict MAU? Please explain why cardiovascular disease diagnosis is not included as a variable. Please define microalbuminuria and pulse wave velocity when they are first mentioned. How well is the result of baPWV correlate with cfpWV?
Author Response
Reviewer: first
Round: first
Manuscript ID: healthcare-580091-original
Manuscript title: Association between Brachial-Ankle Pulse Wave Velocity and Microalbuminuria and to Predict the Risk for the Development of Microalbuminuria Using Brachial-Ankle Pulse Wave Velocity Measurement in Type 2 Diabetes Mellitus Patients
Comments and Suggestions for Authors
General comments:
This study revealed an association between baPWV and MAU and a cut-off value of baPWV to predict MAU in a cohort of T2DM patients. Below are my comments
Comment: The introduction highlights the importance of baPWV and MAU in predicting cardiovascular disease, but it is not clear why it is important to predict MAU using baPWV. Please clarify. MAU can easily be performed in the clinic. What is the reason for using baPWV to predict MAU? Please explain why cardiovascular disease diagnosis is not included as a variable. Please define microalbuminuria and pulse wave velocity when they are first mentioned. How well is the result of baPWV correlate with cfpWV?
Response: Thank you very much for the excellent comments and suggestions. We have attempted to revise the manuscript in a number of places based on reviewer’s comments and the revised contents have been marked with blue colored writings to allow the reviewer’s verifications. We have now clearly mentioned the study need in the introduction section and revised the discussion section- limitations of the study in order to address these reviewer’s comments.
Reviewer 2 Report
In this study, the authors examined the association between baPWV and MAU coupled with predication of MAU using baPWV measurement among Type 2 DM patients in a cross-sectional study. They found that baPWV, cigarette smoking, and LDL-cholesterol were significantly positively associated with MAU in Type 2 DM and suggested that a baPWV cut-off of 1700 cm/sec could be used to predict the presence of MAU in Type 2 DM patients. These findings provide a simple and useful measurement in indicating incipient nephropathy or cardiovascular complications in Type 2 DM, which might be beneficial for the early and appropriate medical intervention among these segments of population.
The manuscript was well written and easy to follow. Subject definitions and methods were described clearly. The topic of this paper was relevant to the field of this journal. I have only a few suggestions for improvement.
Line 48-49: The epidemic information on diabetes mellitus is too old. Please update it to the last 1 to 2 years. The authors suggested a baPWV cut-off of 1700 cm/sec could be used to predict the presence of MAU in Type 2 DM patients. However, the study was based on Korea community. Could the authors provide a short discussion about the applicability of this conclusion in other race such as European and /or American population? Please correct typo errors throughout the manuscript. For example, line 163, correct “Type 2 M” to “Type 2 DM”.
I recommend acceptance of this paper after minor revision.
Author Response
Reviewer: Second
Round: first
Manuscript ID: healthcare-580091-original
Manuscript title: Association between Brachial-Ankle Pulse Wave Velocity and Microalbuminuria and to Predict the Risk for the Development of Microalbuminuria Using Brachial-Ankle Pulse Wave Velocity Measurement in Type 2 Diabetes Mellitus Patients
Comments and Suggestions for Authors
General comments:
In this study, the authors examined the association between baPWV and MAU coupled with predication of MAU using baPWV measurement among Type 2 DM patients in a cross-sectional study. They found that baPWV, cigarette smoking, and LDL-cholesterol were significantly positively associated with MAU in Type 2 DM and suggested that a baPWV cut-off of 1700 cm/sec could be used to predict the presence of MAU in Type 2 DM patients. These findings provide a simple and useful measurement in indicating incipient nephropathy or cardiovascular complications in Type 2 DM, which might be beneficial for the early and appropriate medical intervention among these segments of population.
The manuscript was well written and easy to follow. Subject definitions and methods were described clearly. The topic of this paper was relevant to the field of this journal. I have only a few suggestions for improvement.
Response: Thank you very much for the encouragement. We have revised the manuscript in a number of spaces based on reviewers’ comments and the revised contents have been marked with blue colored writings to allow the reviewers’ verifications.
Comment: Line 48-49: The epidemic information on diabetes mellitus is too old. Please update it to the last 1 to 2 years. The authors suggested a baPWV cut-off of 1700 cm/sec could be used to predict the presence of MAU in Type 2 DM patients. However, the study was based on Korea community. Could the authors provide a short discussion about the applicability of this conclusion in other race such as European and /or American population? Please correct typo errors throughout the manuscript. For example, line 163, correct “Type 2 M” to “Type 2 DM”.
I recommend acceptance of this paper after minor revision.
Response: We highly appreciate the reviewer’s suggestion. We have updated the epidemic information of diabetes mellitus in the introduction section. In addition, predictive value of baPWV for MAU identified in Korean population has been discussed in the limitations of the study for its suitability to generalize this finding. We have thoroughly checked typo errors throughout the manuscript.
Round 2
Reviewer 1 Report
MAU can easily be performed in the clinic. What is the reason for using baPWV to predict MAU? Are there any information in literature regarding how well the result of baPWV correlate with cfpWV?Author Response
Reviewer: first
Round: second
Manuscript ID: healthcare-580091
Manuscript title: Association between Brachial-Ankle Pulse Wave Velocity and Microalbuminuria and to Predict the Risk for the Development of Microalbuminuria Using Brachial-Ankle Pulse Wave Velocity Measurement in Type 2 Diabetes Mellitus Patients
Comments and Suggestions for Authors
General comments:
Minor comments: MAU can easily be performed in the clinic. What is the reason for using baPWV to predict MAU? Are there any information in the literature regarding how will the result of the baPWV correlate with cfpWV?
Response: Thank you very much for some specific comments and suggestions second time. We have attempted to revise these minor comments as suggested. The revised contents have been marked with blue colored writings to allow the reviewer’s verifications.